# Sensing Devices for Detecting and Processing Acoustic Signals in Healthcare

**DOI:** 10.3390/bios12100835

**Published:** 2022-10-07

**Authors:** Norma Mallegni, Giovanna Molinari, Claudio Ricci, Andrea Lazzeri, Davide La Rosa, Antonino Crivello, Mario Milazzo

**Affiliations:** 1Department of Civil and Industrial Engineering, University of Pisa, 56122 Pisa, Italy; 2ISTI-CNR, Institute of Information Science and Technologies, 56124 Pisa, Italy

**Keywords:** healthcare, accelerometers, MEMS, monitoring, acoustic signals

## Abstract

Acoustic signals are important markers to monitor physiological and pathological conditions, e.g., heart and respiratory sounds. The employment of traditional devices, such as stethoscopes, has been progressively superseded by new miniaturized devices, usually identified as microelectromechanical systems (MEMS). These tools are able to better detect the vibrational content of acoustic signals in order to provide a more reliable description of their features (e.g., amplitude, frequency bandwidth). Starting from the description of the structure and working principles of MEMS, we provide a review of their emerging applications in the healthcare field, discussing the advantages and limitations of each framework. Finally, we deliver a discussion on the lessons learned from the literature, and the open questions and challenges in the field that the scientific community must address in the near future.

## 1. Introduction

Vibrations are oscillatory movements close to an equilibrium state that occur in any physical body possessing a mass. Depending on the specific case, such displacements can have a periodic or random nature. Acoustic waves are an example of vibrations that occur through the alternated compression and decompression of a mean (e.g., air, water). They can be perceived as sounds by the hearing apparatus or transient movements of physical bodies (e.g., earthquakes—ground motion) [1].

Acoustic signals in healthcare can be classified into two main groups: external inputs used by auditory apparatus to perceive the surrounding environment, and acoustic manifestations of structural displacements occurring in the body (e.g., heartbeat).

The first class is conveyed to the brain via two mechanisms: bone conduction and air–bone conduction. In the first case, acoustic pressure waves are transferred to the cochlea in the inner ear, and then to the brain, by exploiting the passive conduction given by the bone structure of the skull [2]. In the second case, sounds are collected by the pinna and then transferred to the cochlea passing through the middle ear, composed of the eardrum and the ossicular chain, that filters and transmits acoustic inputs through the synergistic combination of geometries and materials properties of the components [3]. Both these mechanisms, due to pathologies and malfunctions of the native tissues, can be altered, directly affecting the hearing sense [4].

The second class includes, for instance, signals from the cardio-respiratory apparatus, and are usually detected by a stethoscope, a tool that amplifies body sounds through its bell-like shape in contact with the soft tissues closer to the body part being monitored. Although stethoscopes have been widely used by physicians and healthcare operators, their employment presents a few relevant limitations. First, a stethoscope is a piece of equipment that is not wearable but has to be held in close contact with the skin and requires the presence of a trained operator to assess the magnitude and features of the acoustic signal. This also implies a non-neglectable subjectivity in the evaluation of the detected signals that, therefore, does not possess a quantitative nature. Moreover, stethoscopes can be efficiently used for an occasional auscultation but they cannot be employed for continuous and long-term monitoring activities that, in contrast, are required for specific needs (e.g., assessing lung activity during sleep) [5,6].

In order to overcome these issues, microelectromechanical systems (MEMS) have been successfully employed as an alternative to gold standards, to detect and finely measure body vibrations through benchmarks of their dynamics (e.g., displacements, accelerations), or indirectly using other signals (e.g., force). These tools can be properly miniaturized, worn, and customized for each specific application and also for long-term and continuous monitoring activities, eventually serving as point-of-care tools for at-home self-assessments [7].

This review reports the main prototypes and commercial MEMS products for detecting and processing acoustic vibrations in healthcare applications. First, we provide a classification of MEMS based on their working principles and their general employment as reported in the literature. Then, we cover the most studied applications in healthcare where MEMS are used to perceive acoustic signals from environmental or body sources, organizing the dissertation based on the specific targeted organ/physiological phenomenon (Figure 1). A final section is then provided to summarize the take-away messages from the literature and discuss the open challenges to be addressed in the near future.

## 2. MEMS: Classification and Employment

Microelectromechanical systems (MEMS) are mechanical and electro–mechanical elements developed through microfabrication techniques that are usually made of three-dimensional silicon microstructures, ranging between 1 and 100 µm in size. To fabricate such devices, a number of different approaches have been pursued, e.g., film deposition, isotropic and anisotropic etching, and masking and doping techniques [7,8,9].

MEMS are able to convert different types of environmental signals into detectable electrical signals. Specifically, it is possible to highlight six main fields: electrical (e.g., resistance, capacitance, inductance), chemical (e.g., composition, reaction rate, concentration, and pH), mechanical (e.g., displacement, velocity, acceleration, acoustic wavelength, and acoustic intensity), thermal (e.g., temperature, heat), radiative (e.g., intensity, phase, polarization, wavelength), and magnetic (e.g., field intensity, permeability) [10].

MEMS sensors rely on transducers, devices that convert a signal in one form of energy to a signal in another for purposes of measurement or control. Although the word “transducer” has historically been used to refer exclusively to devices that convert mechanical stresses, such as force or pressure into an electrical signal, the current definition has been extended to include all forms of input stimuli (mechanical, electrical, fluidic, or thermal) and output signals other than electrical [11]. The classification of the main types of MEMS sensors, which have been developedis shown in Figure 2.

Inertial sensors are commonly used to estimate body attitudes by measuring linear accelerations and angular rates along the three orthogonal axes; acoustic sensors are used to acquire sounds propagating across different mediums such as air, water, or solids; optical sensors can detect light perturbations in both the visible and invisible spectrum (e.g., IR, UV); radiofrequency sensors are used to acquire electromagnetic signals or perform signal power monitoring; microfluidic sensors are used to handle or process fluids at the microscale level; force sensors are used to detect mechanical pressures; thermopile sensors are used to measure heat fluxes and temperatures with or without contact with the surfaces; chemical sensors are used to detect chemical properties, e.g., gases concentration, pH levels and osmotic pressures.

Thanks to their miniaturization and excellent mechanical and electrical properties, MEMS usually exhibit low power consumption, high sensitivity, light weight, high resolution, stable performance, and ease of integration with other devices and systems. A wide range of MEMS sensors, such as microphones, accelerometers, gyroscopes, pressure, gas, thermal, flow, and biosensors, have been developed and are currently available on the market. Their usage has grown steadily in commercial application, and are currently extensively employed in almost all production fields [12]:*Automotive*, e.g., collision and rollover sensing devices for airbags deployment; fuel level and vapor pressure detection; tire pressure; active suspension and braking monitoring; navigation systems;*Healthcare*, e.g., blood pressure, breathing, glucose and heartbeat sensing; auditory assessment; prosthetics; sleep monitoring; muscle stimulation; drug delivery systems; pacemakers;*Industrial automation*, e.g., machine health monitoring; predictive maintenance; automatic safety mechanisms activation; surveillance of production processes; goods tracking and logistics;*Consumer electronics*, e.g., temperature and vibration monitoring for PC, hard disks, printers, home appliances; gesture recognition for gaming controllers and smartphones; sports training devices;*Environmental and agriculture*, e.g., environmental sensing and weather forecasting, soil fertilization and irrigation planning; crop health monitoring; automated farming;*Telecommunications*, e.g., network devices monitoring; fault detection and localization; electrical and optical signal processing;*Aerospace and defense*, e.g., surveillance; satellite monitoring; UAV and remote system operations; weapons guidance; spacecraft and aircraft navigation.

Independent of the physical phenomena they are designed to acquire, MEMS sensors can be built by exploiting distinct working principles. The main working principles can be classified into eight categories (Figure 3).

Inertial accelerometers, for example, can be manufactured to rely on the piezoelectric, piezoresistive, capacitive or optical effect. Piezoelectric-based sensors exploit the capability of a particular material (usually a metal or a semiconductor) to produce an electrical voltage in response to applied mechanical stress (and vice versa) [13]. Electrothermal sensors are based on materials that generate a voltage difference when exposed to a temperature difference [14]. Resonant sensors are small electromechanical structures that vibrate at high frequencies and typically produce, as the output, a frequency shift induced by the external stimulus altering the mechanical properties of the resonator (e.g., the mass or stiffness) [15]. Electrochemical sensors generate electrical signals in response to chemical reactions [16]. Tunnelling sensors are based on the electron tunnelling gap transducers, which measure a displacement by the change in tunnel current between two electrodes [17]. Capacitive sensors rely on the variation of the electric capacitance of one or more pairs of plates when the distance between them changes due to external stimuli. In contrast, optical sensors are based on manipulating light signals through a mean, e.g., micro mirrors or switches [7]. A different approach involves the so-called “localized surface plasmon resonance” (LSPR), a term that is used to describe the electron density wave that travels over the metal surface. Plasmonic biosensors are a class of devices that use sensitive noble metal nanoparticles (NPs) integrated into a biosensing assembly for applications in environmental pollution analysis, illness diagnosis, and human health monitoring (viral detection) [18,19,20,21].

In healthcare, microphones and accelerometers have been the most commonly employed sensors for acquiring acoustic and vibration signals.

Microphones are acoustic sensors operating in the human audio frequency range (20–20,000 Hz) that are conventionally made of a flexible membrane and a back-plate where a bias voltage is applied. As the sound wave hits the membrane, it induces its oscillation, causing a proportional variation to the electrical capacitance that the coupled electronics can subsequently acquire. Since this model has some limits in terms of maximum signal level and sensitivity to environmental conditions, further sensing mechanisms have been investigated to improve acoustic sensor performance, which include using a back-plate-less design to minimize air damping, using piezoelectric sensing components to achieve low-power directional detection, and optical sensing to deal with extreme environmental conditions. Microphones based on the piezoelectric effect have been predominantly employed due to the simple and robust construction coming from the absence of a backplate, an improved linearity and very low power consumption that allows constant standby [9].

Accelerometers, in contrast, are inertial sensors that can be modeled as a unit composed of a spring, mass and damper, one for each of the sensing directions. The mass is suspended by a specifically designed suspension system that is characterized by a given spring constant. Due to the small size of the micromachined sensing element, an additional viscous damping factor is introduced to dissipate the energy and prevent the mass-spring system from experiencing excessive vibrations. Under an acceleration input, the mass moves relative to a fixed frame structure embedded in the sensor case. Different transduction methods, such as piezoelectric, piezoresistive or capacitive, can be used to measure the displacement of the mass and, hence, determine the inertial force [22]. MEMS accelerometers have advantages over traditional high precision electromechanical sensors, such as small size, extreme ruggedness, low power consumption, and low-cost [23].

Figure 4 shows four typical architectures employed in the design of acoustic and vibration sensors, based on different working mechanisms [24]: in panel (a), the sensing element, which can be a piezoresistive or piezoelectric material, is embedded in an oscillating cantilever holding a suspended mass. The sound waves hitting the structure entail a bending of the sensing element that, in turn, generates a manageable voltage difference. In contrast, in panel (b), the sensing element is assembled onto a diaphragm that deforms with the incoming acoustic waves. The sensing element is made of piezoelectric or piezoresistive materials, or even optical nanofibers, which exploit the propagation interference effect [25]. A capacitive sensing mechanism can also be used by employing both the suspended mass design—panel (c)—or the diaphragm design–panel (d). In such cases, two parallel conducting plates are used: one is fixed to the sensor support structure while the other is free to oscillate. The variable distance among the plates causes a proportional change in their capacitance that is indirectly measured by either controlling the frequency of an oscillator or the attenuation of an alternating current signal.

The architectures based on the suspended mass designs are also commonly used for manufacturing inertial sensors where the primary structure is replicated for each of the axes, along with the forces that are measured.

## 3. Sensing Devices in Healthcare

Sensing devices in healthcare have had, and will have, a huge impact on the future of the healthcare cycle. Currently, clinical devices are able to manage and analyze medical data, environmental conditions, and personal habits from a multitude of (bio)sensors.

In the following subsections, we present the current and most significant applications of sensing devices for detecting and processing acoustic signals in healthcare.

### 3.1. Auditory Apparatus

Hearing loss affects more than 5% of the population worldwide [4]. This condition affects the ability to communicate between individuals, perceive sounds from the surrounding environment, as well as enjoy all leisure activities that involve sound perception, which has been demonstrated to have substantial long impact on the body’s welfare.

Hearing loss can be caused by a malfunction of the middle or the inner ear. In the first case, the condition is named conductive hearing loss since it affects the functioning of the eardrum or the ossicular chain while, in contrast, with sensorineural hearing loss, doctors identify an issue in the cochlea, the part of the inner ear where sound amplification takes place, and pressure waves are converted into electric signals to be transferred to the brain [26].

Conductive hearing loss is usually recovered by intervening with the damaged structure by applying a biological or synthetic tissue replacement to patch/replace the eardrum [27,28] or by using a prosthesis to recover material continuity between the tympanic membrane and the cochlea in place of the damaged ossicles [29,30].

A different situation applies for the sensorineural hearing loss that affects hair cells’ motility in the cochlea, so that they are not able to perform their tasks. In this case, hearing aids for mild/moderate conditions consist of a microphone and a processor that amplifies sounds, delivered directly into the ear canal. However, the amplification is not frequency-sensitive and only 30% of the elderly could benefit from such devices [31].

In the case of relevant conditions, cochlear implants (CIs) are employed to directly stimulate the auditory nerve, based on the amplitudes and frequencies of the external sound, through an array of implanted electrodes, without relying on the hair cells [4,32]. This complex assembly presents external components (microphone and signal processor) connected with the implanted part through a magnetic inductive link. The implanted device is selectively activated by a signal processor that elaborates the sounds perceived by an external microphone [33]. CIs are largely employed, and more than 180,000 people, both children and adults, have benefited from such devices [31].

However, traditional CIs suffer several limitations in terms of day-to-day practicalities (e.g., an external appearance that may induce a social stigma or the impossibility of being treated with a magnetic resonance), and reliability in the long-term [34,35].

Scientists in academia and industry have been developing solutions to improve the design of hearing devices in terms of efficacy and implantability in the body. Specifically, most components, from the battery to the speech processor, can be successfully implanted under the skin close to the pinna. The main issue, for almost total implantability, concerns the tool to detect acoustic signals (e.g., the microphone) [36].

Figure 5 reports a classification of the acoustic sensors for hearing devices able to perceive external acoustic pressures.

The first class is of subcutaneous capacitive sensors that transform the deformation of a membrane into an electric signal. Such devices are usually placed above the pinna to achieve the best directional sensitivity. The first commercial device was presented in 2001, a titanium-based microphone called TICA with dimensions in the order of 4 mm and a weight of 0.4 g, covering a bandwidth up to 10 kHz. Although it was implanted in 20 patients, there are no recent studies on it [37]. A different approach was pursued with TIKI, a tool with two microphones, one external and one subcutaneous, which work in parallel through a microprocessor. In particular, the implanted microphone covers a volume of 7.5 × 28 × 28 mm under the skin, achieving the upper frequency of 6 kHz [38].

Carina™ (Cochlear Ltd., Sydney, Australia) consists of an assembly with a subcutaneous condenser microphone with noise filtering, implanted surgically on the skull behind the subject’s ear. Although its behaviour has been optimized to detect airbone sounds up to 5 kHz and was implanted in 110 patients, successful integration in a totally implanted cochlear implant has not been achieved yet [39,40,41].

Finally, Jung et al. designed a titanium membrane (diameter = 12 mm) with an acoustic titanium tube to increase the first natural frequency. Tests were only conducted in the laboratory using a skin-like membrane made of silicon up to a frequency of 8 kHz [42].

Concerning electromagnetic sensors, Maniglia et al. published an implantable 29 mg displacement sensor comprised of a neodymium-iron-boron magnet encapsulated in a titanium case that was fixed to the malleus. The motion of the malleus allowed the magnet, interacting with a coil fixed on the temporal bone, detect a frequency of up to 3 kHz, creating an electric signal. No clinical tests were reported in literature [43].

An optical sensor based on the reflection of a laser beam on the vibrating tympanic membrane was proposed by Vujanic et al. in 2002. However, it was only a laboratory prototype, and has not been further developed [44].

Piezoresistive MEMS sensors exploiting the acceleration of the incus were developed by Park et al. in 2007. They exploited the low output impedance to enable remote amplification. The prototype covered a volume of 387 × 800 × 230 µm^3^ with a total mass of 166 µm, and consisted of a mass suspended by a flexible beam covered with piezoresistors. Tests were only conducted in laboratory using temporal bones and a laser Doppler vibrometer (LDV) with frequencies up to 7 kHz [45].

Capacitive transduction has also been used for implantable devices [46,47]. A MEMS displacement sensor was developed by Huang et al. in 2007: it consisted of a coiled spring (mass = 15 mg, stiffness = 10 N/m) that transfered the displacement of the umbo to the condenser fixed to the bone walls in a bandwidth up to 5 kHz. The prototype was tested on only one human temporal bone with an LDV [48]. Ko et al., in 2009, improved the previous design, fixing the assembly not on the temporal bone, but directly onto the umbo through springs. The total mass resulted in 25 mg and it was tested in human temporal bones with frequencies up to 8 kHz [49]. Using a capacitance, but exploiting the accelerations instead of the displacements, Zurcher et al. designed a mass plate (14 mg—1 × 1 mm^2^) that moved between fixed walls to generate a voltage. The total weight was 25 mg with a size of 2.5 × 6.2 × 1 mm^3^ and it was fixed on the umbo, catching inputs up to 6 kHz [50]. In 2012, Young et al. investigated the response of a MEMS accelerometer under a vacuum, giving new insights on reduced packaging needs [51]. In contrast, Sachse et al., developed a lumped parameter model of a MEMS capacitive acceleration-based sensor to optimize its features in terms of mechanical and electrical noise, as well as its resonance frequency. A prototype was later fabricated and tested in human temporal bones in a bandwith up to 6 kHz [52].

A different approach was proposed by Woo et al., who aimed to measure pressure variation inside the middle ear cavity due to the vibration of the eardrum. A membrane whose diameter was 10 mm and thickness equal to 20 µm, made of stainless steel, was used as a implanted acoustic sensor in the middle ear cavity [53].

Another exploited physical phenomenon is the piezoelectric effect. Javel et al. published a work in which the acoustic sensor consisted of a piezoelectric bimorph material in the shape of a cantilever beam positioned on the malleus of adult cats. An acousto–mechanical assessment was carried out by measuring the vibration up to 10 kHz a LDV [54].

Esteem^®^ (Envoy Medical Corporation, White Bear Lake, MN, USA), in contrast, is a commercial device that exploits a piezoelectric acoustic sensor to detect the vibration of the middle ear ossicles up to 10 kHz, but requires complex surgery and presents high surgical complication rates [55,56,57].

A different concept was published by Koch et al. who developed a bidirectional membrane transducer fixed at the incudostapedial joint to measure the force passing through the joint. It was made of titanium with a volume of 4 × 2.5 × 1 mm^3^ and a mass equal to 35 mg. Tests were carried out in silico and on a test bench with a reduced bandwidth (0.4–4 kHz) [58,59] (Figure 6).

Kang et al. designed a biocompatible piezoelectric accelerometer using a ceramic bimorph element and an electronic chip enclosed in a titanium case with a total volume of 4.5 × 1 × 0.3 mm^3^ and a mass equal to 38.4 mg. Tests were performed by gluing the device on the incus of a cat and measuring the acoustic stimuli [60]. The same design was tested by Gao et al. with a finite element model that included a human middle ear [61]. Jia et al., in contrast, placed their piezolectric accelerometer on the long process of the incus, achieving an increased volume of up to 5.91 × 2.4 × 2.0 mm^3^ and a higher mass equal to 67 mg. Tests were carried out on seven temporal bones with frequencies up to 10 kHz [62].

Beker et al. changed the goal of using piezoelectric MEMS accelerometers, to using them as CI sensors. They showed a finite element model of the device to harvest energy from the umbo movement, that was later validated in the laboratory using a prototype made of silicone and lead zirconate titanate (PZT) (volume of 4.25 × 4 × 0.525 mm^3^) in a frequency range of 0.5–2.5 kHz [63]. With the same objective, Yip et al. used a piezoelectric MEMS accelerometer made of PZT and validated it with a dedicated amplification circuit on human temporal bones. It was also optimized to reduce the power consumption of the connected cochlear implants in a frequency bandwidth of 0.3-6 kHz. No further technical details were delivered by the authors [64].

More recently, Yüksel et al. presented a similar device that worked in a bandwidth between 0.2 and 5.5 kHz, with dimensions of 5 × 5 × 0.62 mm^3^ (Figure 7). Tests were only carried out on a testbench in the laboratory [65].

A summary of the acoustic sensors for hearing aids, along with the main physical and technical features, is reported in Table 1. As shown, there is a huge variety of sensing devices for the auditory system to best suit the specific condition to treat and, most importantly, because of the different working principles adopted to address the engineering challenge. Therefore, it is difficult to deliver a global analysis in terms on advantages and disadvantages for the different classes. However, it is possible to state that, overall, all the devices aimed to fit the speaking frequency bandwith (1–4 kHz), leaving the highest part of the hearing range (up to 20 kHz) uncovered.

### 3.2. Cardiovascular Apparatus

Bloodstreams make sounds in the frequency range of 20–1000 Hz while flowing through the heart and procuring mechanical displacements [66,67]. Heart sounds are of paramount importance since their features (e.g., intensity, frequency, duration) can be associated with multiple physiological and pathological information related to the health of the heart itself [68,69], and, therefore, have been considered to diagnose and monitor cardiovascular diseases. Human heart sounds are composed of four main components. The two dominant features are the first heart sound (S1-systolic) and the second heart sound (S2-diastolic). The first heart sound, S1, which reproduces the closure of the mitral and the tricuspid valves, is generally distributed in the low-middle frequency range, between 10 Hz to 140 Hz [5]. In contrast, the S2 sound is related to the closure of the aortic and pulmonary valves [70] in a frequency range between 10 Hz and 400 Hz, with an upper bound higher than S1 but, in contrast, with a shorter duration [5]. Additional heart sounds are the third heart sound (S3) and the fourth heart sound (S4) [6] that can be present when patients show congestive heart failure or adventitious pulmonary sounds [71,72] associated with deteriorating cardiopulmonary conditions [72,73,74]. Their frequency distribution is below 50 Hz, and the vibration amplitude is lower than S1 and S2. Due to the weakness of S3 and S4 signals, their identification can be effectively assessed by using highly sensitive accelerometers [71,75,76,77].

Stethoscopes and electrocardiographs (ECGs) have been the most popular tools used to monitor and diagnose heart diseases. However, the main limitation of traditional stethoscopes is related to the high dependency on the clinical experience of doctors, and the limited applicability to long-term medical assessments [5,6]. In contrast, ECGs have been considered more accurate, but there are specific heart diseases that are difficult to diagnose [6].

A method to identify the S1 sound was proposed by Pharm et al. that employed a miniature, battery-operated wearable device as shown in Figure 8. A dedicated algorithm analyzed the power spectral of the acoustic pulse signal to detect the S1 sounds and to remove the artifacts for an accurate heartbeat detection. The results showed a higher accuracy, of up to 98.7%, with an error lower than 0.28 bpm, with respect to a commercial photoplethysmography (PPG) device [78].

In contrast, Sharma et al. developed a wearable device placed on the suprasternal notch at neck. This new device is easier to manage compared with the traditional multiple electrodes attached on the chest, in which the monitored heart sounds are usually corrupted by other external artifacts and from the respiratory cycles [79].

The proposed algorithm was designed to accurately determine the heart rate, avoiding the external noise. Specifically, the heart sound was extracted from the acoustic recording and the cardiac cycle was segmented and classified into S1 and S2 sounds, achieving results with an accuracy of up to 94.34% [80].

In contrast, Chan et al. presented a cuff-less, low cost and ultra-convenient blood pressure monitoring device endowed with a 3-axis accelerometer, positioned on the upper chest. This device was used for the estimation of the systolic and diastolic pressures and was able to monitor the blood pressure at 1 Hz, as long as the accelerations data of the patient was collected and available [81].

A new solution that did not require a direct contact with the patient to control the heartbeat and the heart rate was proposed by Quian et al. The authors employed a microphone and a speaker on a device, e.g., smartphones or laptops. It was possible to generate an acoustic cardiogram (ACG) from inaudible acoustic signals, as schematically depicted in Figure 9. By analyzing the ACG signals obtained from the human body, it was possible to discriminate the heart rate and the heartbeat by employing frequency-modulated sound signals able to discriminate the heart signal from external noise. Moreover, using the double microphone endowed on a mobile device, it was possible to convert the acoustic data into heart and breath rates. Results showed a median heart rate error of 0.6 bpm, and median heartbeat interval error of 19 ms [82].

The use of a MEMS heart sound sensor was investigated by Cut et al., whose novelty consisted of a bionic MEMS based on the pick-up mechanism of the three-dimensional ciliary bundle structure of human ear hair cells. The acoustic sound was analyzed and optimized using analytical and simulation methods, and eventually, experimentally tested. The sensor possessed a small size, and showed very good results such as a high sensitivity to monitor heart sounds (−189.5 dB @ 500 Hz), a good working bandwidth of 10–800 Hz, and low interference with environmental noises [83].

Qu et al., in contrast, developed a device made of a piezoelectric MEMS acoustic sensor with a low-noise amplification circuit integrated into silicone membranes with an air cavity. This light-weight device resulted to be skin compatible, low-cost, and unaffected by external environmental sounds [77].

Numerous cardiovascular diseases are related to abnormal blood pressure values: in particular, about 54% of strokes and 47% of coronary heart diseases are connected to blood pressure, which has become a crucial indicator of a person’s health status [84].

Interestingly, some studies have suggested a correlation between the S2 and the aortic blood pressure [85,86,87,88]. MEMS accelerometers have been used to measure systolic blood pressure [81,89,90], and this approach has represented a good, non-invasive option to monitor patients and predict cardiovascular diseases (e.g., the hypertension disease) [91,92]. Generally speaking, non-invasive diagnoses have relied on the monitoring of the electrical activity or cardiac pump activity, exploiting the dynamic electrocardiogram, phonocardiogram, echocardiogram, or contemporary medical imaging techniques such as NMR, CT, and PET [93].

Among the others, atherosclerosis is a coronary arteries disease that causes the disruption of normal laminar flow of blood and generates stream turbulences. This condition occurs when the walls of the coronary arteries become thicker due to deposited plaques (based on fat, cholesterol, fibers, calcium, and other substances from blood) in the arteries. The accumulation of this plaque restricts the arteria, reducing the normal blood flow, and, consequently, reduces the heart of oxygen. The result is a narrowed lumen in which the blood flow creates a characteristic turbulent sound that can be detected with external tools to avoid hearth attack and failure or arrhythmias [94]. A recent study by Whinter et al. presented an investigation on a large group of patients with low and middle likelihood of coronary artery disease (CAD). The portable acoustic device developed for the detection of CADs was mounted at the fourth intercostal space. The heart sound was analyzed using a dedicated CAD score algorithm that included both acoustic features and clinical risk factors. With a negative predicted value of 96%, this acoustic system could potentially support a clinical assessment, reducing the demand for more advanced and costly diagnostic tools [95]. Table 2 summarizes all the wearable devices developed to monitor heartbeat and blood flow with high accuracy and low cost. Most importantly, they are unsusceptible to external environmental sounds, compared with traditional devices.

### 3.3. Fetus

Accelerometers can be used to monitor fetal heart beats/sounds and movements, picking the signal from the fetus. Acoustic signals from the fetus are sometimes too weak to be detected in the early gestation stage, in particular, before the 30th week of gestation [96]. Fetal body movements are strictly connected with fetal health [97,98,99], and its reduction is frequently a warning of health complications, for example, fetal distress, fetal growth restriction, or hypoxia. Monitoring the fetal phonocardiography is also very important in diagnosing congenital heart disease [96,100]. Currently, the standard methods of fetal monitoring (FM) are limited to their use in clinical environments.

It has been estimated that early neonatal deaths and fresh stillbirths due to birth asphyxia are, respectively, 1 and 1.3 million every year. Urdal et al. studied a multi-crystal strap-on low-cost Doppler device, including an accelerometer, to monitor the fetal heart rate (FHR) during labor using the signals detected by the accelerometer to estimate uterine contractions [101].

Ghosh et al. evaluated the performance of an acoustic sensor-based, cheap, wearable FM monitor that pregnant women could use at home. A thresholding-based signal processing algorithm based on the fusion of the sensors’ outputs to automatically detect FM was developed to analyze the data. The new signal processing algorithm proposed to combine data from all the sensors and to remove artefacts due to maternal movements. The achieved results showed a sensitivity, specificity, and accuracy of 83.3%, 87.8%, and 87.1%, respectively, relative to the maternal sensation of FM [99].

In a later study, Ghosh et al. compared an acoustic sensor, an accelerometer, and a piezoelectric diaphragm as potential candidates for a wearable FM monitoring system. The acoustic sensor and the piezoelectric diaphragm better determined the durations, intensities, and locations of kicks, than the accelerometer. Moreover, they demonstrated that the acoustic sensor and the piezoelectric diaphragm were able to detect weaker fetal movements, compared with the accelerometer [102].

Zakaria et al. developed a system with six accelerometers and an Arduino microcontroller interfaced with a Matlab-based post-processing software to detect fetal movements (Figure 10). The sensors were placed on the maternal abdomen and recorded signals from the fetus, achieving an accuracy of 85.57%, very close to the ultrasound technique, which has an accuracy of about 97%, while the maternal perception is 59.8% [103].

Altini et al. proposed a new wearable device placed on the abdomen to better detect fetal kicks. They proposed a system that combined data from accelerometers and electromyography (EMG). The system drastically reduced the false-positive kick detection [104].

Zhao et al. developed the concept of e-health home-care for fetal signal perception, applying the Internet of Things (IoT) to the system, to connect all the terminal monitoring units to a control center. The wearable system provided four accelerometers for fetal signal acquisition. The signal was processed using a microcontroller via Bluetooth combined with an Android based device, that provided statistics and information on the fetal health status [105]. A schematic of the system is depicted in Figure 11.

Yusenas et al. studied the use of accelerometers and MEMS microphones to develop a device for counting fetal movements to sense various types of fetal movement [106]. In their study, acceleration sensors and MEMS microphones were used to detect three actions performed on the subject’s abdomen: flicking, tapping, and knocking of the fetus. The acceleration sensors showed an accuracy of 69.96% for the tapping action, while the accuracy of the MEMS microphones was 71.11% for the flicking action; however, the accuracy of MEMS microphones was very low for the knocking action (31.11%). The combination of these two devices is a promising tool for monitoring fetal movements.

A good way to prevent fresh stillbirths and early neonatal deaths due to birth asphyxia is by monitoring the regular assessment of the fetal heart rate (FHR) in relation to uterine contractions. Urdal et al. studied a method that reduced the noise increasing the interpretability of FHR Doppler signals, and a method that used accelerometer signals to estimate uterine contractions. The noise in the FHR signal was removed using only the sampled heart rate. Using a three-axis accelerometer set near the Doppler sensor, it was possible to evaluate the contraction when the maternal movement was little [101].

Table 3 lists the novel aspects of fetal monitoring devices that have been proposed to date. These new technologies are able to monitor pregnant women at home without the presence of a doctor and, in particular, show a high accuracy in isolating the mother movements from that of the fetus in addition to when the mother is not in a resting condition.

### 3.4. Respiratory System

Respiratory sounds provide vital information about patients’ health and disease, such as chronic obstructive pulmonary disease (COPD), chronic bronchitis, bronchial asthma, etc. [108].

They are classified as normal or adventitious sounds, whose presence generally indicates a pulmonary disorder [109,110]. The adventitious sounds are of two types: continuous (wheezes and rhonchi) and discontinuous. Crackles, as well, may be produced either by pressure equalization or by a change in elastic stress resulting from the sudden opening of closed airways in the lungs [111].

During the first decades of the 1800s, Rene Theophile Hyacinthe Laennec invented the stethoscope which, since then, has become the most employed tool of every medical setting, and is even considered as synonymous with the profession itself [112].

However, as early as 1985, some scientists [113] pointed out the stethoscope’s limitations because of its low diagnostic value ascribable to the attenuation of higher frequencies which contain valuable diagnostic information regarding respiratory sounds. Indeed, the stethoscope has a frequency response that attenuates frequency components of the lung sound signal above about 120 Hz [114], and the human ear is less sensitive to a lower frequency band. Hence, auscultation through stethoscopes is a subjective process that depends on the physician’s own hearing perception, and the experience and competencies of the medical staff, that can lead to a high inter-observer variability.

Respiratory sound analysis (RSA) or respiratory sound monitoring (RSM) seeks to address the issues of the stethoscope by allowing physicians to record, store and visualize the sounds produced by the respiratory system, as a digital recording, using specific analysis equipment.

The application of new technologies, e.g., electronic stethoscopes, wearable accelerometers, MEMS, could represent a new approach to detecting and diagnosing respiratory disorders [115].

The electronic stethoscope, working as a traditional one, is able to amplify signals to 2000 Hz (respiratory sounds are limited to this frequency) and to record respiratory and heart sounds, which are converted into electrical signals and saved as digital files, for more accurate data processing and transmission [116,117,118].

Moreover, some research groups have used an electronic stethoscope to generate input data for further analysis or classification, using machine learning algorithms such as convolutional neural networks (CNN) or support vector machines (SVM), in the diagnosis of asthma in children [119,120].

Compared with the stethoscope, miniaturized accelerometers can be taped onto the chest wall, integrated into belts, worn on the skin or mounted into clothing, guaranteeing continuous and unobtrusive cardio–respiratory monitoring. Therefore, they can be wearable sensors, as depicted in Figure 12, for different applications and operative scenarios, thus, improving users’ life quality and preventing diseases [121,122].

For instance, asthma is a chronic respiratory disease, whose continuous monitoring becomes relevant for patients’ breathing without missing any asthma attacks. Yuasa et al. [123] proposed continuous breathing monitoring in daily life, using a wearable chest-mounted device, which consisted of a microphone, a photoreceptor and a flexible cover. They was able to show that chest movement could be used for estimating the breathing period and for tracking the asthma attacks of the wearer. In particular, the device was attached to the upper chest with an adhesive gel sheet, while the chest movement signal was acquired by a photoreactor using the deformation of the flexible cover accompanying the respiring chest. In addition, the study allowed estimation of the preferred position for placing the device, because the signal amplitudes at that location were large and less affected by any shoulder movements. A correlation between sound amplitude and tidal volume, detected by a MEMS microphone, was observed. Indeed, the breathing phase identification experiment showed that the periodicity of the chest movements could be used to estimate the breathing periods and phases, whose frequency strongly depends on the wearer’s health condition, such as wheezing, which is typical of an asthma attack [123].

The work of Guesneau et al. [124] evaluated the respiratory rate from the signal of a single-axis accelerometer fixed at the top of the abdomen. The use of a third-order low pass Butterworth filter, the initial estimation of the respiratory rate, turned out to be a deeply accurate method to demonstrate the potential of the accelerometer as a low-cost, non-intrusive method of screening for sleep disorders through respiratory and cardiac signals detection, extracted with a single and in situ measurement.

In the field of respiratory sleep monitoring, the work of Chunhua et al. [125] appeared to be innovative due to the proposal of a novel smart flexible sleep monitoring belt with MEMS triaxial accelerometer, developed to detect vital signs, snore events and sleep stages with an achievable precision of 97.2%. This RS monitoring device was both feasible and effective due to its low cost and high performance with detection accuracies of heart rate and respiration rate of about 1.5 bpm and 0.7 bpm, respectively. Moreover, the sensitivities awake, REM, light sleep, and deep sleep stages were 90.2, 77.1, 78.1 and 73.5%, respectively, results that allow the collection of various vital signs during sleep monitoring.

Similarly, a two-stage amplified PZT sensor was investigated by Chen et al. [126] for monitoring lung and heart sounds in discharged pneumonia patients. In the study, they used a self-developed sound sensor based on a novel asymmetric gapped cantilever composed of a piezoelectric beam made of piezoelectric ceramic, to continuously monitor lung and heart sounds. The idea was to convert the biomechanical energy (such as the acoustic vibration) to electric energy due to the piezoelectric effect, reaching high sensitivity at a frequency less than 1000 Hz, suitable for weak lung and heart sound monitoring, gaining great potential for clinical use and home-use health monitoring.

The use of piezoelectric devices was also exploited within the investigation of Nguyen et al. [127], where a MEMS-based microphone and a piezoresistive cantilever were able to measure a 0.1-mPa acoustic signal with a frequency down to 2 Hz. The obtained highly sensitive low-frequency device showed a compliance that was 200 times higher than the conventional piezoresistive cantilever, with an SNR of ~80 dB in the range of 2 to 200 Hz, optimum in different applications, such as healthcare and photoacoustic-based gas/chemical sensing, etc.

The novel coronavirus disease (COVID-19), which has rapidly swept around the globe, has created the need for technological solutions for medical-preventive actions, such as the capability to continuously monitor key physiological parameters of the disease.

In particular, the study of Xiaoyue et al. [128] proposed an automated wireless device, tailored for COVID-19 patients, able to detect vital signs and respiratory activity, such as cough, in revealing the early signs of infection and in quantitating the responses to therapeutics, both in clinical and home settings. The system was characterized by soft, skin-mounted electronics that incorporated high-bandwidth and a miniaturized motion sensor, enabling digital and wireless measurements of mechano–acoustic (MA) signatures of both core vital signs (i.e., heart rate, respiratory rate and temperature) and underexplored biomarkers (coughing count), as indicators of both disease and infectiousness (Figure 13).

Similarly, Qixin et al. [129] proposed the idea of a triboelectric nanogenerator for respiratory sensing (RS-TENG), which was designed and integrated with a facemask that endowed respiratory monitoring function due to small volume, easy fabrication, simple installation and economical applicability, helpful for developing multifunctional health monitoring gadgets during the COVID-19 pandemic.

Finally, the employment of smartphones for recording respiratory sounds has gained relevant interest [130,131]. Indeed, the built-in microphone was found to be a low-cost, contact-free, trustable, and straightforward solution for breathing monitoring, even though more studies and clinical validation are still required [115].

For instance, smartphones could be used for wheeze recognition using an SVM classifier in pediatric patients [132], or for developing an automatic system to detect crackle sounds [133].

In [132], the presented device was 71.4% accurate in the sensitivity and 88.9% in the specificity of the recorded sounds, requiring no direct contact with the patients, no standardized environment, and only a standard smartphone. Conversely, in [133], the authors described a device that allowed crackle detection with an accuracy ranging from 84.86 to 89.16%, a sensitivity ranging from 93.45 to 97.65%, and a specificity ranging from 99.82 to 99.84%. Moreover, it led to successful results related to crackle disclosure, the most masked noise among the frequency components of respiratory sounds.

More recently, an app was developed for this purpose, taking 15 s to detect a crackling sound [134] using a smartphone’s microphone to assess a potential COVID-19 infection [130]. Table 4 summarizes and highlights some of the most innovative features related to the proposed devices for RS monitoring, to date. Such sensing typologies appear rather contextualized within the current scenario, which is strongly marked by the recent pandemic emergency. Concerning this specific scenario, such devices also allow a continuous monitoring of the patient without a clinician present.

### 3.5. Gastrointestinal Tract

Many people suffer from motility and functional bowel disorders that require an effective assessment of their intestinal conditions, playing a vital role in the diagnosis and evaluation of eventual diseases [136]. Indeed, out of nearly eight billion human beings, it is likely that almost all emit and/or have heard bowel sounds (BS) [137].

Although such sounds are closely linked to vital processes of life and health, they are notoriously difficult to be directly measured since they occur randomly in time and location with very low amplitudes, compared with other body sounds. As a matter of fact, many physicians initially proceed with invasive testing (blood tests, stool tests, colonoscopy and/or biopsy) to rule out potentially fatal organic disease before confirming the less serious diagnosis. These invasive tests are not only unpleasant for patients, but could carry significant risks together with physical discomfort, psychological distress and financial costs due to time off work.

In addition, there is a diffused lack of effective tools other than the non-invasive practice of auscultation [138]. However, while the sounds from the lungs and heart have been widely investigated due to their characteristic and regular patterns, capturing sounds produced by the stomach and intestines remains an open challenge.

Hence, for decades, many doctors and researchers have developed miniaturized, easy-to-fabricate and wearable medical devices to support medical diagnosis as well as to reduce cost [139].

Mamun et al. [136] proposed a novel, ultra-low power, real time bowel sound detector, able to measure meal instances in artificial pancreas devices. This system provided aid to long-term diabetic patients by the use of a front-end detector that transduced the initial bowel sound, recorded from a piezoelectric sensor, into a voltage signal. Therefore, it provided a non-invasive approach to detect and to correlate physiological measures in real time with motility or meal instances. This system only consumed 53 μW of power and was implemented on a 0.96 mm^2^ chip space. The frequency of bowel is well below 500 Hz, so the detecting systems should detect bowel sounds with high accuracy in the presence of environmental noise. The sensitivity of the proposed implant was not only easily tailored, but could also isolate abdominal vibrations from a noise spectrum signal dominated by the heartbeat, or from noise from talking and walking, showing about 85% accuracy in detection of gastrointestinal sounds with a low number of false-positives.

In contrast, Dagdeviren et al. [140] showed the importance of piezoelectric-based devices in detecting bowel sounds. They reported and designed an ingestible, flexible piezoelectric device that sensed mechanical deformation within the gastric cavity in both in vitro and ex vivo simulated gastric models. Indeed, despite advances in device development for GI monitoring, significant risks associated with solid, non-flexible gastrointestinal transiting systems remain, that can eventually lead to intestinal obstruction or are related to limited battery lifespan drawbacks. This device does not display cytotoxicity for cell metabolic activity or for plasma membrane integrity. Furthermore, cell adhesion and spread was observed over the device surface, highlighting its potential biocompatibility. Concerning the electrical performance, the PZT GI-S behavior depended on the type of bowel motion occurring during the assessment. For instance, when air was introduced into the stomach, the device reported a pressure increase given indirectly by a significant voltage increase from about 10 to 60 mV. Then, after 40 s of inflation, pressure appeared to be stabilized (plateau voltage curve) while, after the air was released, a decrease in voltage occurred. The proposed example showed the potential sensitivity of the device to detect changes associated with air ingestion, as well as the capacity to guide evaluation and treatment in cases of aerophagia or intestinal bacterial overgrowth.

Hence, the development of a system capable of both sensing and remaining flexible within the gastrointestinal (GI) environment, eventually reducing the mentioned risks, may have a good impact on the diagnosis and treatment of motility disorders. The small dimensions and flexible nature of such a device could also reduce the likelihood of GI tissue damage, maximizing its broad applicability. Similarly, the smart shirt for digestion acoustics monitoring, named GastroDigitalShirt, implemented by Baronetto et al. [141], monitored the different digestion phases (peristaltic contractions) across six hours in participants with no prior GI diseases, capturing the main bowel sound (BS) types reported in the literature. The prototype embedded an array of eight miniaturized microphones connected to a low-powered wearable computer for performing long-term, automated auscultation to clinically monitor digestion and track a number of specific disease symptoms.

In the field of wearable devices, the work of Fengle et al. has gained great interest. They developed a flexible, skin-mounted device for long-term and real-time monitoring/evaluation of bowel sounds based on the integration of a three-dimensional printed elastomeric resonator with flexible electronics, attached to abdominal surfaces. Clinical tests, conducted on patients with mechanical intestinal obstruction or paralytic ileus, highlighted the relevance of the device for capturing the characteristics of bowel sounds as an auxiliary tool in the diagnosis of bowel issues [139].

Another flexible device, useful for the digital auscultation of bowel sound monitoring, was proposed by Gang et al. [142]. They produced a flexible dual-channel with active noise reduction, wireless, wearable and conformably attached to the abdominal skin. It allowed the continuous wearable monitoring of BSs for patients with postoperative ileus (POI) from pre-operation (POD0) to postoperative day 7 (POD7), providing performant guidance for doctors to choose a reasonable feeding time for patients after surgery and to accelerate their recovery. The main innovation of the abovementioned soft, light, and thin device was the ability of digital auscultation with active noise reduction due to a synchronous acquisition channel for ambient noise. The adaptive filter was used to subtract the ambient noise from the noise-contaminated BS signals, with performant results for BS monitoring in noisy clinical environments. The maximum NRR of active noise reduction was −19.7 dB in testing under a sound level of 45 dB of ambient noise, rendering the detected final frequency closer to the common standard value of doctors’ auscultation. The presented device used a bandpass filter to elaborate low-frequency internal noise that possessed a bandwidth of 10–20 Hz. In contrast, after the filter, the only frequency peak was at 280 Hz, corresponding to the potential peak in the BS spectrum. However, the traditional bandpass filter was not able to effectively suppress ambient noise. In attempting to solve this further issue, Wang et al. adopted an adaptive filter, able to optimize the subtraction between the ambient noise contained in BS signals [142].

Table 5 summarizes the keener and more innovative devices in BS detection, not only for gastrointestinal motility sensing, but also for continuous wearable monitoring in postoperative ileus patients. Such devices are a solid alternative to the traditional invasive screening and prevention procedures. In fact, they are characterized, for instance, by ingestible devices, free of any type of cytotoxicity, and with high biocompatibility within the gastrointestinal system.

### 3.6. Sleep Monitoring

According to the National Institute of Health, sleep is an integral part of the daily human routine, as essential as food or water. Therefore, several efforts have been made to promote research on sleep and related areas. Sleep is a physiological process repeated every 24 h, and its description has involved researchers, clinicians, physiologists, technicians, and engineers [143].

Currently, it is generally accepted that the essential sleep variables to characterize the sleep process are: the amount of time needed to accomplish the transition from awake to sleep (i.e., sleep onset latency, SOL), the total amount of sleep (i.e., total sleep time, TST), the amount of wake time in minutes during the sleeping period after sleep onset has been achieved (e.g., wake after sleep onset, WASO), the sleep efficiency (SE) commonly defined as the ratio of TST and time in bed, and the number of awakenings (NWAK) during the night [144]. This consensus starts from the definition of the cyclical pattern of sleep, composed of a rapid eye movement (REM) and non-REM (NREM) phase. The NREM phase is generally divided into four different stages, namely, Stage 1, Stage 2, Stage 3, and Stage 4. Knowledge of these stages allows the further inference of new variables. In a clinical setting, the gold-standard device to characterize human sleep is considered polysomnography (PSG). PSG is a diagnostic tool able to record several physiologic parameters and signals through electroencephalography, electrocardiography, oximetry, and measures of respiration generally exploiting chest belts [145].

PSG is used to infer the different sleep stages, and represents an indirect measure of sleep. Unfortunately, it is expensive, and a medical doctor generally supervises the tool in a hospital setting. Consequently, many alternatives have been proposed in the literature to enable sleep monitoring in a less intrusive manner and to allow continuous monitoring at home.

Recent advances in smart health, both from the hardware and software point of view, have, therefore, led to solutions for clinical settings and home monitoring. Wearable technologies can monitor movements and physiological parameters (e.g., heart rate and respiratory rate). In this context, actigraphy represents a typical wearable sleep monitoring device: sleep and wake phases are detected by gathering information from body movements, typically using sensors worn on the wrist [146]. Through inertial sensors, typically accelerometer, gyroscope, and magnetometer, these devices are also able to estimate orientation, and acceleration, thus, inferring information about body movements [147]. The medical device industry have, thus, presented many devices both for sleep monitoring and sleep apnea detection. These devices have relied on different combinations of raw signals.

From a clinical standpoint, many devices have been approved by national public health administrations. In addition to the above mentioned actigraphy and polysomnography, which are widely used in clinical and hospital settings, a few wearable devices have been marked as “medical device” or have obtained FDA clearance. For example, the AcuPebble SA100 is a small and wearable device able to detect obstrusive sleep apneas in adults through the sensing of physiological sounds generated by the body and the detection of respiratory and cardiac biomarkers. Acoustic signals are transferred to a mobile device, and finally to the cloud, for further processing. The device achieves good accuracy compared with gold standard devices [148]. Another interesting device that has received FDA clerance is the DROWZLE Sleep Apnea, a mobile application that claims to be able to detect sleep apneas using the patient’s phone [149].

From the consumer point of view, several proposals have been presented as enabling technologies to monitor sleep through the heart rate: ballistocardiography, piezoelectric, and photoplethysmography [150]. As an example, Emfit QS [151] is a ballistocardiography-based device that exploits a thin strip placed under the mattress which is able to evaluate sleep stages, giving information about apnea or bed occupancy. Similarly, the Withings sleep device exploits a strip underneath the mattress and provides sleep stages, snoring information, and HR monitoring during the night. In contrast, the Beddit device allows the identification of similar features but relies on a no-contact piezoeletric strip [152]. Another devise is the Beautyrest, which exploits passive piezoelectric sensors to detect human pressure on the mattress [153]. Finally, an interesting new technology applied to sleep medicine is the photoplethysmography (PPG). PPG, similar to a pulse oximeter, exploits a light source and photodetector to identity differences of light intensity caused by vascular tissue and blood flow. Based on PPG, remarkable consumer devices are produced by Garmin, Fitbit, Whithing, and Xiaomi, and are able to identify sleep stages through heart rate monitoring [154]. PPG is also used to evaluate respiratory rate and, in the market, the Oura ring represents a consumer device able to identify sleep stages and total sleep time [155]. It is important to remark that sleep monitoring devices show important differences in sleep stage evaluation when compared with outputs of gold-standard devices. However, as a main drawback, commercial devices are not able to identify sleep disorders.

Figure 14 shows a summary of recently developed systems with applications in sleep monitoring.

Wearable devices also incorporate technology to monitor the heart rate, but generally are not able to accurately discriminate between sleep stages. Typically, they can discriminate if a user is awake, asleep, sleep duration and time awake. Validation studies about actigraphy and wearable devices have shown an agreement with PSG of typically around 85–90% [156,157]. The literature shows that the accuracy of such devices has increased year after year, and the performance of commercial products has started to be comparable with gold-standard devices for sleep monitoring. In this review, we are focusing on sensing devices for processing acoustic signals, and, therefore, we summarize in Table 6 those MEMS-based wearable devices also able to collect heart rate and/or respiratory information. Furthermore, we analyze recent findings on devices utilized to detect obstructive sleep apnea syndromes (OSAs), often simply called apneas, namely, a breathing disorder characterized by temporary obstructions of the upper airways (no breathing episodes) during sleep [158]. In this regard, the authors of a recent review reported the most relevant studies on the analysis of acoustic properties for classifying OSAs including peak intensity, duration, and occurrences [159].

Table 6 shows some of the innovative approaches in the wearable devices industry applied to the sleep framework. Despite some of the mentioned devices having reached a research-grade prototype, the Table shows an increasing interest in sleep monitoring, especially in the home setting. It is interesting to note that HR was the most-used information, and that researchers tried to include respiratory information using accelerometers or piezoelectric sensors. Another aspect to note, is the importance of collecting and publishing public and open datasets of human sleep sessions, and more effort should be spent in comparing devices’ outputs with the gold-standard devices generally used by clinicians.

In order to avoid wearable or intrusive devices, researchers have also proposed technologies to enable sleep tracking in a non-instrusive but objective manner. Generally, these devices are not classified as medical devices, and their main target is to understand user habits. As well shown in [165], unobstrusive sleep monitoring is mainly performed through the analysis of cardiac, breathing, and moving events. Authors also offer a taxonomy of existing unobstrusive methods for sleep assessment, as shown in Figure 15.

In this context, MEMSs are particularly useful for detecting acoustic signals from the heart and, moreover, the analysis of the sound during breathing cycles could lead to understanding apnea events.

The emerging technologies based on MEMS include ballistocardiography, ultrasounds, and phonocardiographic sensors. Ballistocardiography built upon accelerometers, relies on the representation of cardiac events in order to give insights on blood injections. Ballistocragraphy based on accelerometers, can infer about cardiac events by detecting the induced vibrations. In contrast, ultrasound sensors have been used to process breathing rate and body movement. In [164], the authors proposed a no-contact ultrasonic device to quantify breathing activity. Based on a low power ultrasonic active source and transducer, the device measured the frequency shift produced by the velocity difference between the exhaled air flow and the ambient environment, i.e., the Doppler effect.

Existing methods and studies have shown remarkable effort in developing new methods to assess sleep information, nevertheless, they are generally not able to correctly identify all the sleep stages, i.e., REM, SWS, and NREM1-3. In this regard, wearable systems are more suitable for enabling home sleep monitoring. Recent advancements, new sensing materials, and advancements in data analysis and deep learning methods could lead to the full assessment of human sleep in non clinical settings, allowing a wide spread of tools for monitoring and preventing sleep problems in the population.

Table 7 shows a few innovative examples of non-obtrusive and innovative devices for sleep monitoring. These devices mainly rely on microphones in order to understand breathing or snoring. The most interesting approach is represented by the ballistocardiograph applied to sleep monitoring. In addition to interesting results recently reached by this technology, a wide validation campaign seems missing and, furthermore, these devices seem strongly influenced by several external factors, e.g., the mattress thickness and users’ weight.

Finally, an interesting topic strictly related to sleep monitoring is lucid dreaming [169]. It is the phenomenon that occurs when a person is aware that they are dreaming, and can influence dreaming thoughts.

Lucid dreams arise most frequently during REM sleep, but rarely during NREM or immediately after the awake state. According to [170], currently the main challenge in this field is to develop a reliable system able to induce this phenomenon. In fact, lucid dreaming is rare, but the capability of inducing it could have a huge clinical application, e.g., for treating recurrent nightmares in post-traumatic stress disorder [171]. As a consequence, industries have started to introduce devices into the market that claim to induce lucid dreaming. The authors of [170] offered an interesting review of the most recent devices for inducing lucid dreaming. Generally, these devices rely on electroencephalographic activity and eye movement. A few devices available in the market are also equipped with MEMS and accelerometers. For example, Aurora is a headband equipped with electrodes for EEG and accelerometers for detecting body movements [172]. Another interesting example is the Hypnodyne’s ZMax device. Similar to Aurora, it is a headband able to emit light, it is vibrotactile, and emits auditory stimuli. The device collects sleep information through frontal sensors able to detect brain activity and ocular movements, in addition to the collection of heart rate, temperature, sound, and body movements through accelerometers [170]. Although lucid dreaming applications have not directly exploited MEMS and accelerometers, as for sleep and apnea monitoring, they could represent a plus for developing more accurate and reliable assessments in sleep medicine.

## 4. Conclusions and Future Overlook

MEMS have revolutionized data collection in many fields, but healthcare is definitely one of the frameworks in which they have had a major impact. In particular, the possibility of reliably catching information from physiological acoustic signals has changed the approach of monitoring health conditions and providing personalized treatments to patients. The employment of MEMS has progressively changed the traditional approach consisting of using stethoscopes, to qualitatively appreciate the small vibrational signals from internal organs or in relation to auditory apparatus, have improved the capability of sensing devices to transmit pressure variations to improve the hearing sense and, more importantly, the quality of life in patients with hearing loss conditions.

Figure 16 shows a roadmap of miniaturized sensing devices in healthcare, highlighting past, present, and future directions. In the next five to ten years, we could witness a wider range of applications, from implantable sensors to reliable vital signals monitoring and telemetry.

Despite the high number of useful benefits and opportunities for the near future, the employment of miniaturized systems in “acoustic” healthcare has not really taken off, because of a few limitations. First, acoustic signals generally possess small amplitudes that are easily affected by different sources of noise: from noises from mechanical actions, e.g., friction between the device and the hosting tissue/fabric, to the more common electromagnetic noises due to external aleatory environmental conditions, particularly relevant in long-term monitoring procedures. Moreover, wearable devices are not always accepted for use on a daily basis, because they may limit movements and, psychologically, may cause discomfort.

Therefore, from the technical standpoint, there a several challenges to be addressed. The first challenge concerns the importance of lowering not only the footprint, but also the cost of the entire life-cycle of the products, from the energy costs for the fabrication, to costs related to the final end-users and, eventually, for their potential recycling. From a design standpoint, effort for the development of a new generation of nanoscale sensing materials is envisaged, aiming also to achieve a synergistic employment of different, and occasionally overlooked, materials [173].

Moreover, the future of sensing devices in healthcare is also related to efficient algorithms able to efficiently analyze big data and signals, involving users’ personalization and clinical experts’ guidelines. A personalized healthcare will combine continuous health monitoring, and real-time feedback from end-users and clinicians.

As reported, being an interdisciplinary field, we envisage a stronger collaboration and involvement of different professional figures, including data scientists and engineers that can deliver new techniques to collect and process data, including artificial intelligence, thus, giving devices the capability of being “smart” in terms of data acquisition, processing, and visualization in real time and remotely, in the global framework of the so-called Internet of Things. This approach, particularly relevant in the current Industry 4.0 scenario, will further improve the conditions and life quality of patients, making easier, at the same time, the periodical clinical checks from professionals. The involvement of (bio)materials experts will also improve the physical and psychological conditions of subjects, limiting issues with internal hosting tissues, as well as creating “smart” fabrics able to better comply with the body, thus, reducing the discomfort of wearing sensing devices.

Moreover, new research avenues could be considered, especially those that have been only marginally treated by the scientific community, including sounds from swallowing for dysphagic patients [174], or acoustic electromyography [175].

In addition to the technical aspects, clinicians struggle to fully trust new devices and have preferred using the golden standards (e.g., stethoscopes, large pieces of equipment), even though they are often costlier and more invasive. Therefore, in the near future, an improved trust of MEMS must be built in order to increase their use in current clinics. This will enable closer contact between the scientific community and the end users, while also allowing the collection of more data that will help improve current designs to overcome current technical limitations.

## Figures and Tables

**Figure 1 biosensors-12-00835-f001:**
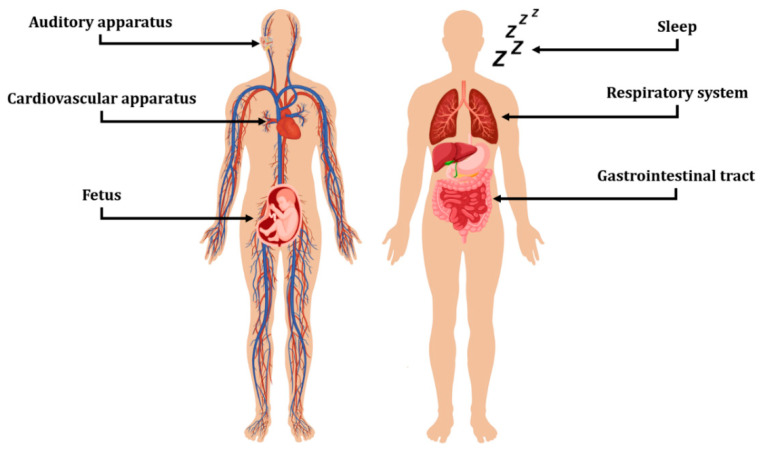
Fields of application for acoustic sensing devices covered in this work. The Figure was assembled using images from freepik.com and Vecteezy.com.

**Figure 2 biosensors-12-00835-f002:**
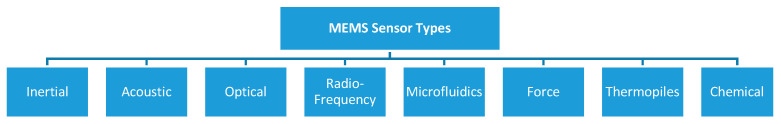
Classification of MEMS sensor types.

**Figure 3 biosensors-12-00835-f003:**
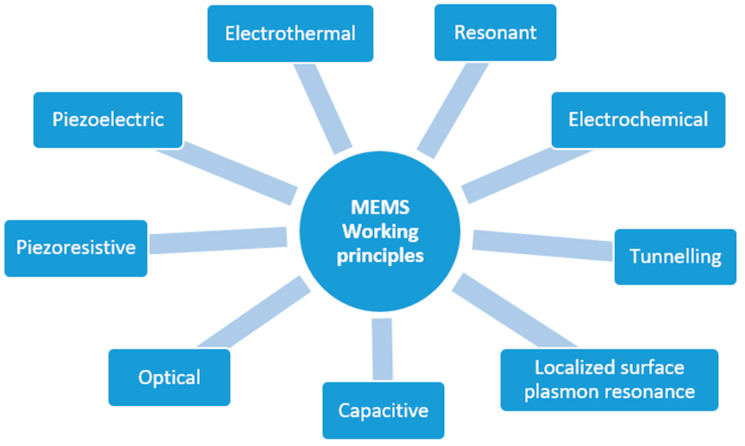
Working principles of MEMS sensors.

**Figure 4 biosensors-12-00835-f004:**
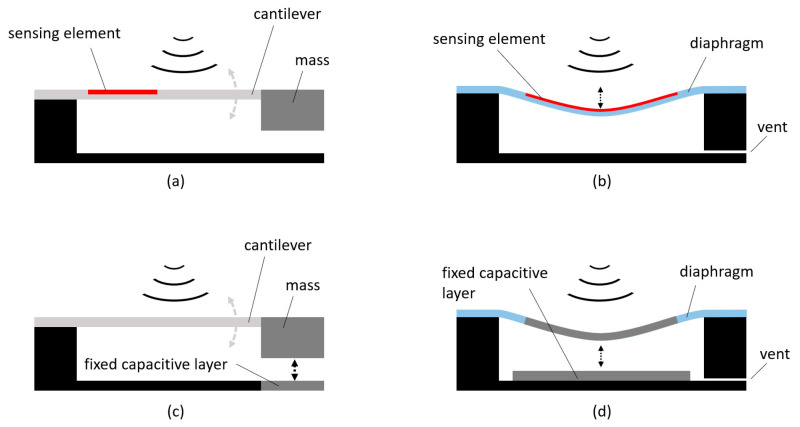
Common architectures of acoustic and vibration sensors. Panel (**a**) cantilever with embedded sensing element and suspended mass. Panel (**b**) diaphragm with embedded sensing element. Panel (**c**) cantilever with capacitive layers and suspended mass. Panel (**d**) diaphragm with capacitive layers.

**Figure 5 biosensors-12-00835-f005:**
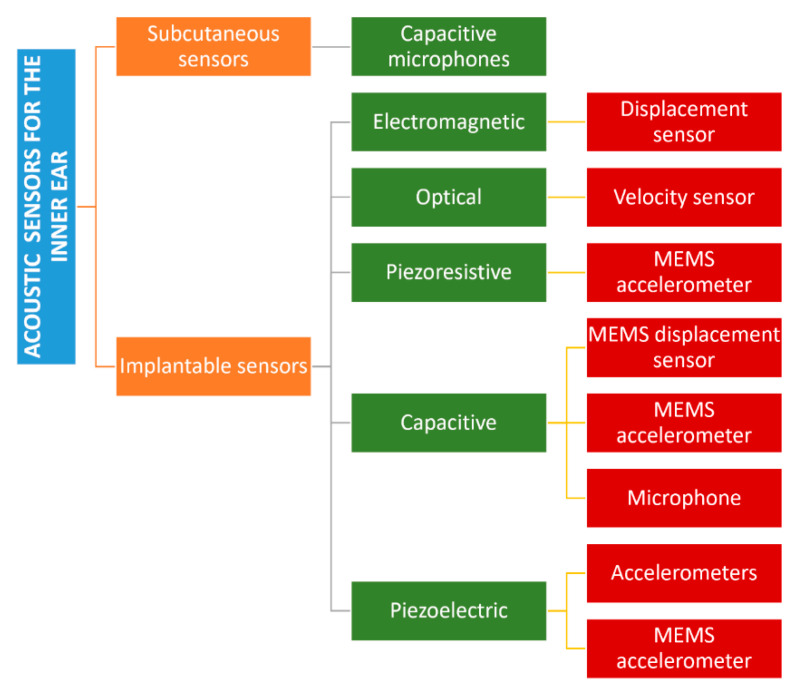
Classification of the acoustic sensors for hearing aids based on the implantation site and the final application.

**Figure 6 biosensors-12-00835-f006:**
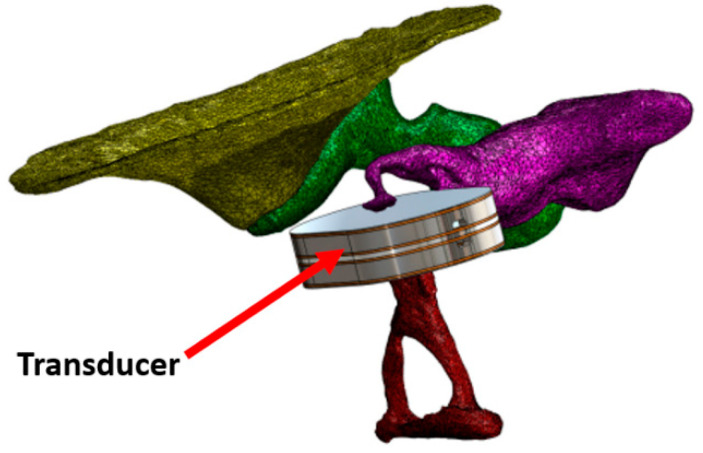
Transducer to estimate the force passing through the incudostapedial joint. Open Access [59] © 2014 MDPI.

**Figure 7 biosensors-12-00835-f007:**
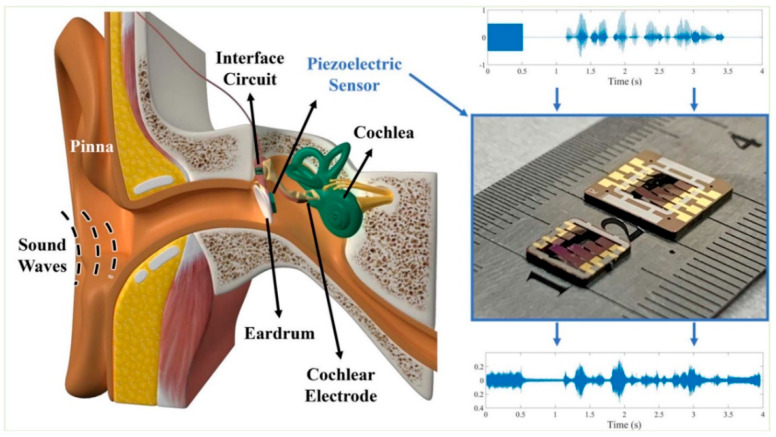
Piezoelectric MEMS sensor to feed cochlear electrodes. Open Access [65] © 2021 IEEE.

**Figure 8 biosensors-12-00835-f008:**
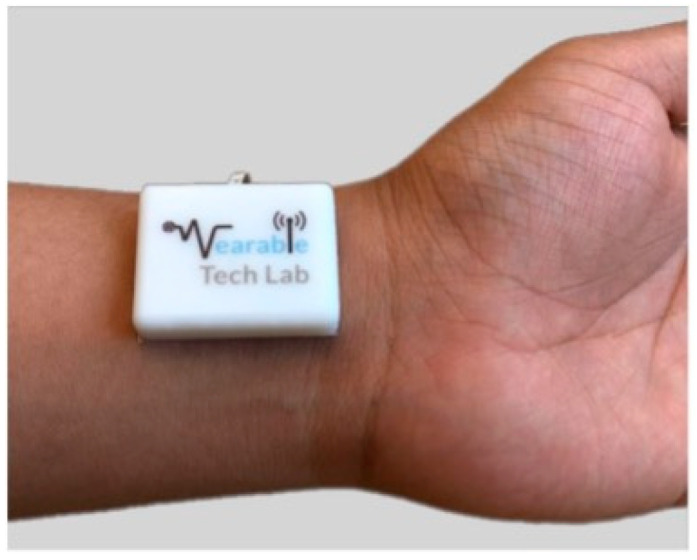
Wearable device to monitor the heart sound at the wrist. The device is composed of a MEMS microphone sensor integrated with Bluetooth. Open Access [78] © 2019 Nature Publishing Group.

**Figure 9 biosensors-12-00835-f009:**
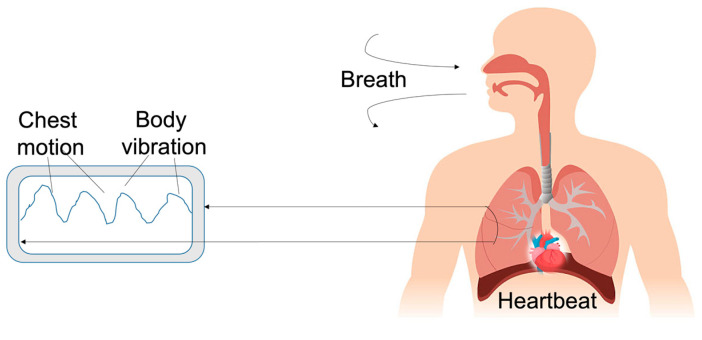
Schematic of the acoustic cardiogram using inaudible acoustic signals to monitor heartbeat.

**Figure 10 biosensors-12-00835-f010:**
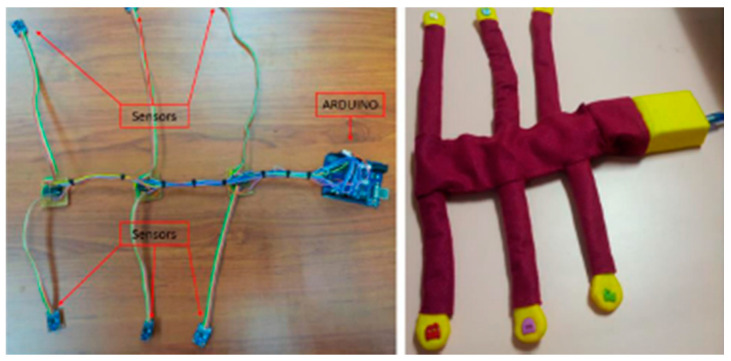
The recording system with six accelerometer sensors and an ARDUINO microcontroller. Open Access [103] © 2018 ARPN—Asian Research Publishing Network.

**Figure 11 biosensors-12-00835-f011:**
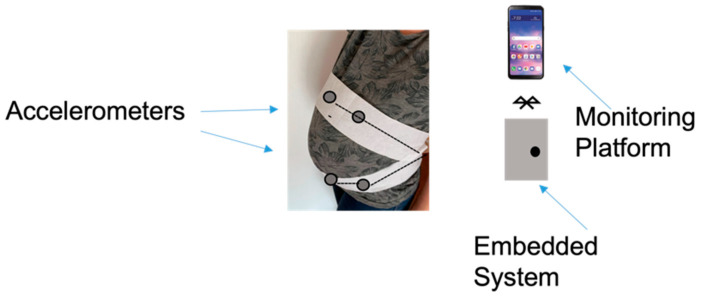
Components of the local fetal movement monitoring device.

**Figure 12 biosensors-12-00835-f012:**
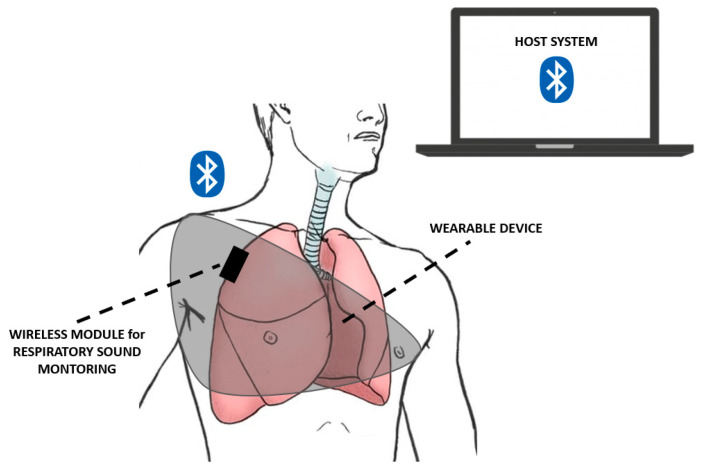
Schematics of a wearable and wireless breathing sound monitoring system as presented in [108].

**Figure 13 biosensors-12-00835-f013:**
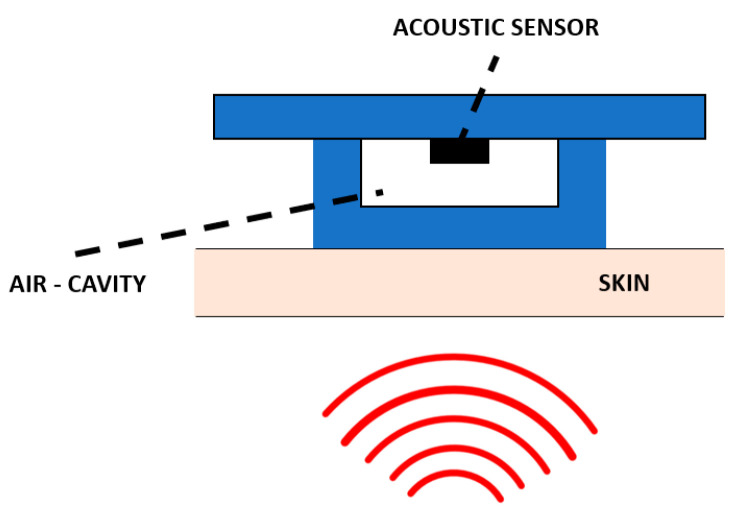
Working principle of the air–silicone composite device as presented in [128].

**Figure 14 biosensors-12-00835-f014:**
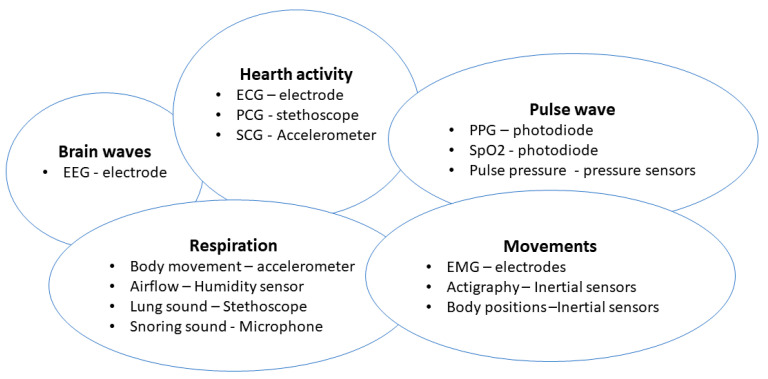
Classification of the most recent systems and applications for sleep monitor.

**Figure 15 biosensors-12-00835-f015:**
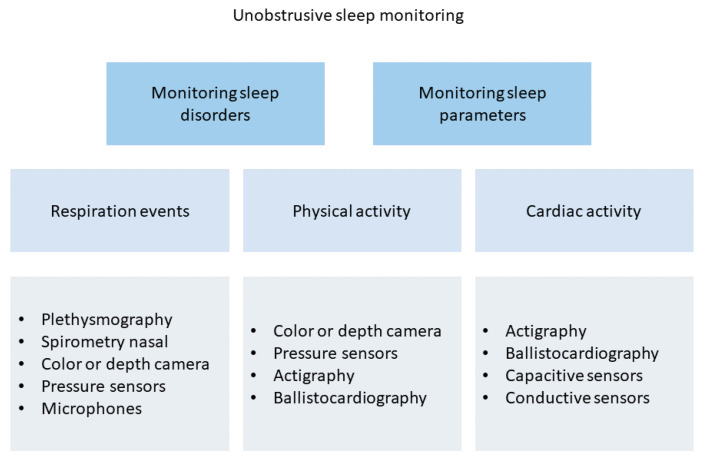
Taxonomy of existing unobstrusive methods for sleep assessment.

**Figure 16 biosensors-12-00835-f016:**
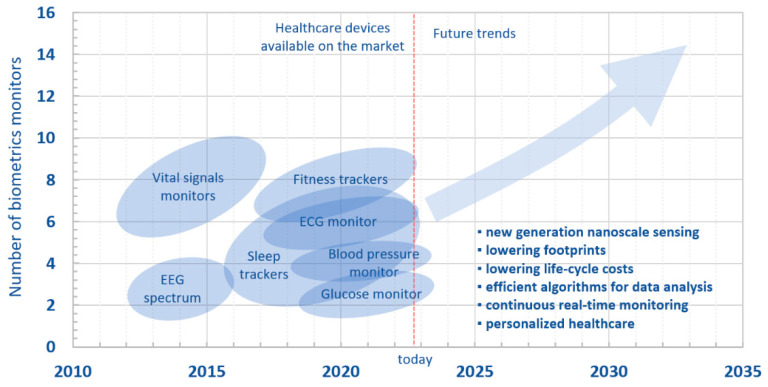
Roadmap of sensing devices for healthcare.

**Table 1 biosensors-12-00835-t001:** Acoustic sensors for hearing aids.

Device	Size	Weight	Bandwidth	Sensitivity	Power Consumption	Tests	Ref.
TICA: subcutaneous microphone based on a capacitive membrane	Diameter: 4.5 mm	Mass: 0.4 g	0.1–10 kHz	5 dB ref. 1 mV/Pa	0.05–0.5 mW	Implanted in 20 patients	[37]
TIKI: two capacitive microphones (one subcutaneous and one external)	Volume: 7.5 × 28 × 28 mm^3^	^-^	0.2–6 kHz	10 dB ref. 1 mV/Pa	0.05–0.5 mW	Implanted in 3 patients	[38]
Carina™: subcutaneous microphone with condenser microphone	-	-	0.25–5 kHz	-	0.05–0.5 mW	Implanted in 110 patients, but a full integration was not achieved	[40]
Subcutaneous microphone with a titanium membrane	Diameter: 12 mm	-	0.1–8 kHz	35 dB ref. 1 mV/Pa	0.05–0.5 mW	Tested in the laboratory with a silicone-made skin	[42]
Electromagnetic sensor: interaction between a magnet fixed on the malleus and a fixed coil	-	-	0.25–3 kHz	30 dB ref. 1 mV/Pa	≈1 mW	Tested in the laboratory	[43]
Optical sensor based on the reflection of a laser beam on the tympanic membrane (or one of the ossicles)	-	-	0.5–10 kHz	46 dB ref. 1 mV/Pa	6.4 mW	Tested in the laboratory	[44]
Piezoresistive MEMS that measured the acceleration of the incus	Volume: 387 × 800 × 230 µm^3^	-	0.9–7 kHz	46 dB ref. 1 mV/Pa	>1 mW	Tested in the laboratory with temporal bones and the LDV	[45]
Capacitive MEMS displacement sensor based on a coiled spring that transferred the displacement of the umbo to the condenser	-	-	0.5–5 kHz	20 dB ref. 1 mV/Pa	≈4.5 mW	Tested on one temporal bone and the LDV	[48]
Capacitive MEMS displacement sensor fixed on the umbo through springs	-	25 mg	0.8–8 kHz	20 dB ref. 1 mV/Pa	≈4.5 mW	Tested on temporal bones	[49]
Capacitive MEMS acceleration sensor: a plate moved between fixed walls to generate capacitance-related voltage	Volume: 2.5 × 6.2 × 1 mm^3^	25 mg	0.2–6 kHz	19 dB ref. 1 mV/Pa	≈4.5 mW	Tested in the laboratory	[50]
Capacitive MEMS acceleration sensor	-	-	0.5–6 kHz	9 dB ref. 1 mV/Pa	-	Optimization through modeling; tested in the laboratory with human temporal bones	[52]
Capacitive MEMS that measured the middle ear pressure variation due to the motion of the eardrum	Diameter: 10 mm; thickness: 20 µm	-	0.1–10 kHz	28 dB ref. 1 mV/Pa	≈1 mW	Tested in the laboratory	[53]
Piezoelectric bimorph material in the shape of a cantilever beam positioned on the malleus of adult cats	-	-	0.5–10 kHz	45 dB ref. 1 mV/Pa	-	An acousto–mechanical assessment was carried out by measuring the vibration of the sensor with an LDV	[54]
Esteem^®^: a piezoelectric acoustic sensor to detect the vibration of the middle ear ossicles	-	-	0.25–8 kHz	-	-	Tested on 134 patients	[55,56,57]
Piezoelectric force sensor: a bidirectional membrane transducer fixed at the incudostapedial joint to measure the force passing through the joint	Volume: 4 × 2.5 × 1 mm^3^	35 mg	0.25–8 kHz	-	-	Tests in silico and on a test bench	[58,59]
Piezoelectric accelerometer sensor: made of a ceramic bimorph element and an electronic chip enclosed in a titanium case	Volume: 4.5 × 1 × 0.3 mm^3^	38.4 mg	0.4–4 kHz	15 dB ref. 1 mV/Pa	-	Tests in the laboratory on the incus of cats and with a finite element model that included the human middle ear	[60,61]
Piezoelectric accelerometer sensor: placed on the long process of the incus	Volume:	67 mg	0.4–4 kHz	15 dB ref. 1 mV/Pa	≈1 mW	Tested on seven temporal bones	[62]
Piezoelectric accelerometer sensor: to harvest energy from the umbo movement for powering cochlear implants	-	-	0.5–2.5 kHz	62 dB ref. 1 mV/Pa	-	Finite element model and tested in the laboratory	[63]
Piezoelectric (PZT) accelerometer sensor: for powering cochlear implants	-	-	0.3–6 kHz	20 dB ref. 1 mV/Pa	0.01 mW	Tested on temporal bones to minimize energy consumption	[64]
Piezoelectric (PZT) accelerometer sensor: for powering cochlear implants	Volume: 5 × 5 × 0.62 mm^3^	4.8 mg	0.25–5.5 kHz	26 dB ref. 1 mV/Pa	-	Tested in the laboratory	[65]

**Table 2 biosensors-12-00835-t002:** Devices to detect heart-related acoustic signals.

Device	Mechanism	Application	Results	Ref.
New miniature, battery-operated wearable device	Wearable device to monitor the heart rate at the wrist	An algorithm analyzed the acoustic pulse signal to detect S1 sounds	Removed the artifacts for an accurate heartbeat detection, with an accuracy of 98.7% and an error lower than 0.28 bpm, compared with a commercial photoplethysmography (ppg) device	[78]
Wearable device	Wearable device placed on the suprasternal notch at neck	The algorithm determined the heart rate, avoiding the external noise	The results showed an accuracy of 94.34% for the heart rate determination	[80]
Portable acoustic device	The device was mounted at the fourth intercostal space	Dedicated CAD score algorithm that included both acoustic features and clinical risk factors	A negative predicted value of 96%, this device could reduce the demand for more advanced and costly diagnostic tools	[95]
Device for blood pressure monitoring	The device was endowed with a 3-axis accelerometer, positioned on the upper chest	Estimated the systolic and diastolic pressures and monitored the blood pressure	Low cost and cuff-less, able to monitor the blood pressure at 1 Hz	[81]
Non-contact device to control the heartbeat and the heart rate	A microphone and a speaker on a device, e.g., smartphones or laptops	ACG discriminated the heart rate and the heartbeat by using frequency-modulated sound signals to identify the heart signal from the external noise	Results showed a median heart rate error of 0.6 bpm, and median heartbeat interval error of 19 ms	[82]
Bionic MEMS	Device based on the pick-up mechanism of the three-dimensional ciliary bundle structure of human ear hair cells	The acoustic sound was analyzed using analytical and simulation methods, and experimental test	Small size, high sensitivity to monitor the heart sounds—189.5 db @ 500 Hz, a bandwidth 10–800 Hz, and low interference with environmental noises	[83]
Heart sound sensor attached on the chest	Piezoelectric MEMS acoustic sensor: a low noise amplification circuit, and silicone polymers	The device recorded different heart sounds while at rest and after training activities	Low cost, light-weight, skin compatible device, unsusceptible to external environmental sounds; good stability	[77]

**Table 3 biosensors-12-00835-t003:** Devices to detect fetal acoustic signals.

Device	Mechanism	Application	Result	Ref.
A cheap acoustic sensor-based device used by pregnant women at home	Wearable acoustic sensor monitor	Analysis of fetal movements	Developed a thresholding-based signal processing algorithm to detect fetal sounds by removing artefacts due to maternal movements	[99]
A comparative study of an acoustic sensor, accelerometer, and piezoelectric diaphragm as candidate vibration sensors for a wearable FM monitor	A silicon-based membrane similar to the abdomen to mimic the vibrations due to fetal kicks	Captured fetal sounds produced by the kicks	Better determined the durations, intensities, and locations of fetal kick movements	[102]
Six accelerometer sensors and ARDUINO microcontroller	MATLAB signal process tool to record, display and store relevant fetal movement	The sensors were placed on the maternal abdomen to record and process the signals from the fetal	Performed better in identifying episodes of fetal activity and episodes of inactivity	[103]
IoT-based wearable system for fetal movement monitoring using accelerometers and machine learning	Internet of Things (IoT) applied on the system to connect all terminal monitoring units to a control center; the system consisted of two parts: the local monitoring unit and the remote health evaluation unit	Local monitoring unit	E- health home care, Internet of Things (IoT) was applied on the system to connect all the terminal monitoring units to a control center	[105]
Acceleration sensors and MEMS microphones	The devices were used to detect three actions performed on the subject’s abdomen: flicking, tapping, and knocking	The noise was removed using the sampled heart rate; the three-axes accelerometer set near the Doppler sensor, evaluated the contraction	The accuracy of using acceleration sensors was 69.96% for the tapping action; while for MEMS microphones was 71.11% for the flicking action	[106]
A multi-crystal strap-on low-cost Doppler device, including an accelerometer	Developed methods to increase FHR Doppler signals by reducing noise and estimating uterine contractions using accelerometers	-	Noise present in the FHR signal was reduced, for good detection of contractions when the maternal movement was low	[101]
Wearable system	Combination of accelerometers and bespoke acoustic	Local monitoring unit on the maternal abdomen	Successfully discriminated fetal and maternal movements, and also movements when the mother was active	[107]
Single wearable system	Variable length accelerometer, combination data with electromyography	Single wearable device placed on the abdomen	Decreased false-positive kick detection, and separation from the maternal noise	[106]

**Table 4 biosensors-12-00835-t004:** Sensor equipment for respiratory sound monitoring (RSM).

Device	Mechanism	Application	Result	Ref.
Hybrid-based aspiration and respiration sensing (HARS)	Elastic flexible cover, microphone (MEMS) and photoreactor	Monitoring of asthma attacks and detection of the breathing phases	Breathing phase identification of patient condition, such as wheezing	[123]
Single-axis accelerometer accessorized with a third-order low pass Butterworth filter	Wearable elastic belt, worn around the subject’s abdomen	Non-intrusive method of screening for sleep disorders and patient follow-up	Testing of three different breathing modes: normal, slow and fast, with an accuracy of breathing frequency evaluation <1%	[124]
Digital signal processor (DSP) circuit and a flexible sensor film	Belt packaged by the micro-fiber cloth of a PET-flexible sensor film consisting of a pressure sensor array and a MEMS triaxial accelerometer	Detection of heart and respiration rates, snoring recognition, and sleep stages classification	Accuracies of heart rate and respiration rate reached by the belt were about 1.5 bpm and 0.7 bpm, respectively. Accuracy of 97.2% in the snoring recognition method	[125]
Small-sized and ultra-sensitive accelerometer	A sound sensor made of an asymmetric-gapped cantilever structure with a ceramic piezoelectric beam in zirconium titanate (PZT) as the top layer	Lung and heart sound monitoring in discharged pneumonia patients	Tracking of the recovery course of pneumonia patients with a rapid, simple and highly sensitive detection of lung and heart sounds with a great potential for clinical use and home-use health monitoring	[126]
MEMS-based microphone	Sensing piezoresistive cantilever with ultra-high acoustic compliance	Applications in healthcare, monitoring	Small size device with a SNR of ~80 dB in the range of 2 to 200 Hz and a high SNR in low-frequency range	[127]
MA sensors	Wearable real-time monitoring system	Monitoring of COVID-19 infections due to a continuous record of coughing frequency and intensity	Detection of decay trend of coughing frequency and intensity through the course of disease recovery	[128]
Intelligent facemask	An ultrathin FEP film and Al foil as triboelectric layers and a conductive cloth tape as electrode	Respiratory sensing (RS-TENG) for coronavirus disease (COVID-19), integrated with facemask	High sensitivity and feasibility in respiratory monitoring in diagnosing many respiratory diseases of human bodies, able to give timely alarm after breathing stops	[129]
Built-in microphone of a smartphone	Recording, by a built-in microphone and the headset microphone of an iPhone 4S (Apple, Inc., Cupertino, CA, USA) placed on the subject’s neck and nose	Detecting nasal airflow and tracheal breath sounds	Accuracy with median errors of less than 1% for the nasal sound, for all breathing ranges even if the smartphone’s microphone was as far as 30 cm away from the nose	[130]
Microphone of a smartphone	Smartphone (SH-12C, Sharp Corp., Osaka, Japan) attached to the anterior chest wall over the sternum using adhesive tape	Detection of snoring condition as a marker of obstructive sleep apnea (OSA) and vascular risk	Diagnostic sensitivity and specificity of 0.70 and 0.94, respectively	[131]
Microphone of a smartphone	Microphone (SP0410HR5H-PB, distributed by Knowles Electronics, Itasca, IL, USA) of the Nexus 4™ smartphone (Google, Mountain View, CA, USA) placed near the mouth	Monitoring of wheezing respiratory diseases in children as an outpatient objective tool for recognition of wheezing	Achieved 71.4% sensitivity and 88.9% specificity in wheezing detection	[132]
Smartphone-based system	Electret subminiature microphone (BT-2159000, Knowles Electronics, Itasca, IL, USA) encapsulated in a plastic bell and a smartphone device containing the developed mobile app governing sounds acquisition, display and processing	Recording, storage and analysis of respiratory sounds with crackle detection	Accuracy ranged from 84.86 to 89.16%, sensitivity ranged from 93.45 to 97.65%, and specificity ranged from 99.82 to 99.84%	[133,134]
Piezoelectric MEMS acoustic sensor	A wearable mechano–acoustic sensing device consisting of a piezoelectric MEMS acoustic sensor, a low noise amplification circuit and silicone polymers for the package	Monitoring of heart sound and detection of speech and voice	Light-weight, low cost and skin-compatible device to enable mechano–acoustic sensing including heartbeat and speaking	[135]

**Table 5 biosensors-12-00835-t005:** Sensor equipment for gastrointestinal sound monitoring.

Device	Mechanism	Application	Result	Ref.
Integrated real time bowel sound detector	Wearable piezoelectric sensor	The physiological measure of meal instances in artificial pancreas devices	Remote real-time monitoring was achievable with wireless technologies in an easy-to-fabricate, low-cost, light-weight and wearable device based on a piezoelectric MEMS acoustic sensor	[136]
Flexible printed circuit board (fPCB)	Polyimide fPCB film, equipped with two auscultation and an overall structure of silicone packaging	Bowel sound monitoring	Continuous wearable monitoring of BS for patients with postoperative ileus (POI) from pre-operation (POD0) to postoperative day 7 (POD7)	[140]
Smart Shirt	Slim-fit T-shirt as a substrate for the microphone system, in elastane, and with an embedded microphone matrix	The capture of abdominal sounds produced during digestion	Substantial data collected with the accurate detection of 4 BS types, as reported in the literature	[141]
Flexible piezoelectric devices	A PZT GI-S encapsulated with a 1.2 µm-thick layer of polyimide and a 10 µm-thick layer of ultraviolet curable epoxy (LOCTITE 5055; Henkel), equipped with an electrical connection and computer-controllable USB multimeter	Gastrointestinal motility sensing	The ingestible device, sensing mechanical deformation within the gastric cavity, was able to quantify the behaviors of the gastrointestinal tract using computational modelling	[142]

**Table 6 biosensors-12-00835-t006:** Wearable devices for sleep monitoring and detection of OSAs.

Device	Mechanism	Application	Result	Ref.
SimpleLink multi-standard sensorTag CC2650STK which includes an inertial measuring unit (IMU)	An HR monitoring system that uses the angular rate data from a single axis of a MEMS gyroscope to detect heartbeats. IMU was secured to the chest using an elastic fabric belt	Detection of HR in real-time for sleep physicians	Real-time monitoring, but method still required further testing with a larger pool of participants in real-life scenarios	[160]
Apple Watch	Apple Watch with a mobile application was applied to the wrist, containing a digital sleep diary and psychomotor vigilance test, sending data remotely in real-time. Mobile application needed for accessing the accelerometer and heart rate data was in the Apple Watch	Distinguished sleep from wake, and determined sleep stages, compared with gold-standard PSG	The sleep/wake classification (without motion) was consistent. However, the wake/NREM/REM neural net classifier achieved a best accuracy of 69%	[161]
Patch system composed of ECG, stethoscope, impedance, 9-axial magnetic, angular rate and gravity (MARG) sensors, a digital stethoscope and ambient sound recording	Using the derived standard ECG leads, authors classified the ECG fiducial points and peaks in the stethoscope signal. Respiratory signals and rates were estimated using ECG-derived respiration techniques combined with a novel phonocardiogram-derived respiration approach	Estimation of heart rate and heart rate variability, detailed ECG activity, and respiratory monitoring	Application on long-term home sleep monitoring	[162]
MEMS-based 3-axis accelerometer with digital output	Accelerometer placed on the space between the 7th rib and the region above the diaphragm (the solar plexus). If apnea was detected, a signal was sent to the wristband, and a vibration started until the patient started breathing again	Detection of apnea events in real-time and alerting the patient	A study group of 10 patients who had sleep apnea (SA). All apnea events were detected, and all patients were successfully alerted	[163]
Surface acoustic wave (SAW) sensor (piezo-electric based)	The flexible and passive sensor was placed around the nostrils, monitoring respiration using its sensitivity to humidity change	Nasal airflow monitoring and sleep apnea detection	SAW sensors were suitable for OSAS monitoring with good sensitivity, reliability, and response time	[164]

**Table 7 biosensors-12-00835-t007:** Unobtrusive sleep monitoring solutions which exploit acoustic signals.

Device	Mechanism	Application	Result	Ref.
Embletta^®^X100 PSG: IoT sleep tracking platform, including ballistocardiography, environmental sensors, and actigraphy.	BCG sensor used was the SCA11H bed sensor from Murata. Environmental data loggers, including humidity, temperature, light and sound.	Portable and long-term sleep monitoring	The model could distinguish between sleep and wake states, but could not classify each sleep stage as accurately as PSG	[166]
Ultrasonic transducers	A 40 kHz ultrasound transmitter illuminated an area including the subject’s head. One receiver, tuned to the same frequency, recovered the signal reflected from the scene.	Breath monitoring with a focus on sleep apnea detection	A validation on a subject equipped with a pressure sensor connected to a nasal cannula showed the synchronization of the pressure signal provided by the nasal cannula with the signal spectrogram.	[167]
Microphone	The reflections from the human body arrived at a specific time depending on the distance from the phone speaker. Focusing on the corresponding frequency allowed authors to reliably extract the amplitude changes due to breathing.	Detection of apneas and estimation of total sleep time	Results from a clinical study with 37 patients showed good performances on detecting apnea events. The apneas–hypopneas index could be improved. Other respiratory-related events or physiological information were not detected.	[168]

## Data Availability

Not applicable.

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
