# Peer review of "Sensing Devices for Detecting and Processing Acoustic Signals in Healthcare"

_biosensors, 2022, doi:10.3390/bios12100835_

Round 1

Reviewer 1 Report

1/It is suggested that main types of MEMS sensor in fig.2 used in industry should be including chemical sensors for example gas sensors.

2/It is recommended that the table 1 header names be redesigned to easy on comparison. The technical features are subdivided into bandwidth, sensitivity, power consumption, size and weight etc.

3/It is suggested to quote some typical  pictures of acoustic  sensors structure to facilitate comparison and understand the innovation by reader.

4/It is suggested to make a summary for each table, and indicate the advantages or disadvantages of the results and the practical applications feasibility etc..

5/It is suggested to review that what challenges should be addressed about the sensing devices in the future except AI.

Author Response

Please consider the attached file.

Reviewer 2 Report

The manuscript biosensors-1917255 corresponds to a review related to the study of particular sensing devices for detecting and processing acoustic signals in healthcare. Please see below a list of comments to the authors:

  1. A roadmap of representative sensing devices in this topic would be welcome.
  2. Considering that this work has been submitted to the special issue “Micro Biosensing Systems for Healthcare Applications” Why acoustic based biosensors are not part of this review?
  3. The authors are invited to consider some perspectives about the potential of “Nano- and Micro-Technologies in Biosensors” for instance in acousto-plasmonic sensing. You can see for instance: doi:10.3390/mi8110321 and https://doi.org/10.1364/OE.455595
  4. Only in particular cases of the text are described aspects about the noise or noise to signal ration in acoustic signals. General aspects could be described in a subsection.
  5. Please comment about the sensitivity in the systems presented for table 5.
  6. Figure 2 requires a citation or a better description and justification.
  7. Figure 4 requires a citation or a better description and justification.
  8. Only one reference related to machine learning was analyzed for this topic. Why?
  9. It is suggested to split the collective citations in order to better justify the references selected for this review with individual expressions.
  10. The keywords should be separated by semicolon.

Author Response

Please consider the attached file.

Reviewer 3 Report

The article is well written. A few things to consider in this paper are as below:

1. In line 387, the authors do not give any examples of cardiovascular diseases. Can the authors include examples? 

2. In figure 4, the classification of the acoustic sensors needs to be revised. The information included does not seem accurate, as the optical sensors do not rely on the velocity sensors, and neither the displacement sensors on the electromagnetism.

3. Can the authors include more information on sleeping monitoring/sleep apnea devices? As these devices are extensively used in the medical device industry.  

4. Can the authors provide examples of companies that manufacture such devices? If not all, at least those approved by the FDA.

5. Can the authors include techniques used in detecting cardiovascular diseases using non-invasive methods?

6. Lucid dreaming is another field that authors have excluded in this study. There are a bunch of medical device companies working on this using MEMS system and would be good to include in this paper.

Author Response

Please consider the attached file.

Round 2

Reviewer 2 Report

The manuscript is original and it has been well prepared. The information presented can be useful for future research. The reviewed version of the report has been improved and then I can recommend this work for publication in present form.

Reviewer 3 Report

Dear authors,

Thank you for addressing the feedback provided. The article is well presented with a natural flow to the readers.